# Molecular Design of Novel Protein-Degrading Therapeutics Agents Currently in Clinical Trial

**DOI:** 10.3390/pharmaceutics17060744

**Published:** 2025-06-05

**Authors:** Ela Kacin, Raj Nayan Sewduth

**Affiliations:** VIB-KU Leuven Center for Cancer Biology (VIB), 3000 Leuven, Belgium

**Keywords:** PROTACs, peptide drug design, peptide drugs, targeted protein degradation, multi-omics, micropeptides, peptide drug delivery

## Abstract

The landscape of clinical trials aimed at targeting specific proteins has experienced significant advancements, presenting promising opportunities for the development of effective therapeutics across a range of diseases. These trials focus on the investigation of modulation of protein functions, utilizing innovative technologies such as PROTACs (Proteolysis-Targeting Chimeras) and other protein degraders. These innovative approaches aim to address previously undruggable targets, enhancing the specificity and efficacy of treatments. The current landscape of clinical trials encompasses a diverse array of therapeutic areas, including oncology, autoimmune diseases, and neurological disorders. For instance, drugs like ARV-471 and ARV-110 are in advanced phases for treating metastatic breast cancer and prostate cancer, respectively, by targeting estrogen and androgen receptors. Early-phase trials explored the potential of targeting proteins like IKZF1/3 in multiple myeloma and IRAK4 in autoimmune diseases. The conducted trials not only emphasize the therapeutic potential of protein degradation but also highlight the challenges associated with bioavailability, stability, and delivery mechanisms. As these clinical trials advance, they possess the potential to transform treatment paradigms, providing renewed hope for patients facing complex and refractory conditions.

## 1. Introduction

The advancement of targeted therapies has significantly transformed the medical landscape, providing renewed optimism for the treatment of various diseases through the precise targeting of fundamental molecular mechanisms. Among these cutting-edge strategies, PROTACs (Proteolysis-Targeting Chimeras) and other protein degraders have surfaced as a highly promising approach. These therapies function by selectively degrading proteins that contribute to disease pathology, rather than simply inhibiting their activity, which can result in more effective and sustainable treatment outcomes.

PROTACs and analogous technologies leverage the inherent protein degradation pathways of cells, such as the ubiquitin–proteasome system, to selectively eliminate specific proteins. This approach offers numerous advantages over conventional therapies, including the capacity to target proteins previously deemed “undruggable” and to address challenges associated with drug resistance. Consequently, these innovative therapies are being investigated in a diverse array of clinical trials, focusing on conditions that range from cancer to autoimmune diseases and neurological disorders.

The current landscape of clinical trials reflects the growing interest and investment in this area. For instance, drugs like ARV-471 and ARV-110 are in advanced stages of clinical testing for breast and prostate cancers, respectively, demonstrating the potential of PROTACs to address significant unmet medical needs. Early-phase trials are also investigating the efficacy of targeting proteins involved in multiple myeloma, autoimmune diseases, and other complex conditions (Figure 1).

Despite the promising results, challenges remain to be addressed, including the optimization of bioavailability and stability of these compounds, as well as ensuring their safe and effective delivery to target tissues. Ongoing research and clinical trials are continuously refining these therapies, with the aim of translating their potential into tangible clinical benefits. In summary, the targeted degradation of specific proteins represents a cutting-edge approach in therapeutic development, possessing the potential to transform the treatment landscape for numerous diseases. As clinical trials advance, they provide valuable insights into the efficacy and safety of these innovative therapies, paving the way for future advancements in precision medicine.

While a growing number of reviews focus on the molecular engineering of targeted protein degradation, the rapidly evolving clinical landscape and the emergence of novel degradation modalities necessitate an updated and comprehensive analysis. Our review stands out by providing the most recent clinical trial data on PROTACs and other protein degraders, critically evaluating molecular design strategies, and highlighting translational challenges unique to this therapeutic class. We also incorporate a comparative discussion of alternative technologies, such as LYTACs and molecular glues, which are only briefly mentioned in prior reviews. By integrating these elements, our work offers readers a timely and distinctive perspective that bridges the gap between molecular innovation and clinical application in protein degradation therapeutics

## 2. Main Text

All data presented in this review were collected from peer-reviewed publications, clinical trial registries (e.g., ClinicalTrials.gov), and company press releases as of 15th of May 2025. Detailed references are provided for each clinical candidate and mechanistic insight discussed, ensuring transparency and reproducibility of the information presented.

### 2.1. Current Status of Clinical Trials Against PROTACs

Table 1 provides an overview of various clinical trials involving PROTACs and other targeted protein degraders. These small molecules represent a novel therapeutic approach by selectively degrading disease-associated proteins via the ubiquitin–proteasome system, thereby eliminating their pathological activity rather than merely inhibiting their function [1,2]. The table summarizes key drug candidates, detailing their molecular targets, indications, clinical trial phases, and the companies developing them. Among the most advanced candidates, ARV-471, developed by Arvinas and Pfizer, is currently in Phase 3 clinical trials for metastatic breast cancer [3]. ARV-471 targets the estrogen receptor (ER), a key driver of hormone receptor-positive (HR+) breast cancer. Unlike conventional selective estrogen receptor degraders (SERDs), such as fulvestrant, ARV-471 induces more complete ER degradation, offering a potential advantage in hormone therapy-resistant patients [3].

Similarly, ARV-110, another PROTAC developed by Arvinas, is in Phase 2 trials for prostate cancer [4]. ARV-110 selectively degrades the androgen receptor (AR), which is a critical driver of prostate cancer progression. This drug is being investigated particularly in patients with castration-resistant prostate cancer (CRPC), a condition in which tumors become resistant to androgen receptor inhibitors like enzalutamide and abiraterone. By depleting AR protein levels, ARV-110 offers a promising approach to overcoming treatment resistance [4]. In the field of hematologic malignancies, several PROTACs are under evaluation. CFT7455, a IKZF1/3 degrader developed by C4 Therapeutics, is in Phase 1 trials for relapsed/refractory multiple myeloma (RRMM) [5]. This PROTAC selectively degrades Ikaros (IKZF1) and Aiolos (IKZF3), two essential transcription factors in multiple myeloma cell survival. Unlike immunomodulatory drugs (IMiDs) such as lenalidomide, which modulate IKZF1/3 activity, CFT7455 leads to complete degradation, which may enhance therapeutic efficacy in patients with IMiD-resistant disease [5].

Beyond oncology, PROTACs are also being explored in autoimmune diseases. KT-474, developed by Kymera Therapeutics, is a Phase 1 PROTAC targeting IRAK4, a kinase involved in Toll-like receptor (TLR) and IL-1 receptor (IL-1R) signaling [6]. IRAK4 plays a pivotal role in inflammatory responses, and its degradation represents a novel mechanism for controlling autoimmune and inflammatory diseases such as rheumatoid arthritis and hidradenitis suppurativa. Preliminary data suggest that KT-474 effectively suppresses pro-inflammatory cytokine production, making it a potential alternative to traditional immunosuppressive therapies [6]. Several other PROTACs are being investigated for B-cell malignancies and lymphomas. NX-2127, developed by Nurix Therapeutics, degrades Bruton’s Tyrosine Kinase (BTK) and is currently in Phase 1 trials for B-cell malignancies, including chronic lymphocytic leukemia (CLL) and mantle cell lymphoma (MCL) [7]. Unlike conventional BTK inhibitors, such as ibrutinib, which often fail due to resistance mutations in BTK, NX-2127 induces complete BTK degradation, offering a potential treatment strategy for patients resistant to standard therapies [7]. Additionally, KT-413, another PROTAC developed by Kymera Therapeutics, is in Phase 1 trials for lymphomas, targeting IRAK4 and IMiD substrates, further expanding the therapeutic landscape for hematologic malignancies [8].

In the area of solid tumors, CFT8634, a BRD9 degrader by C4 Therapeutics, is being evaluated in Phase 1 trials for synovial sarcoma, a rare and aggressive soft tissue malignancy [9]. BRD9 is a component of the SWI/SNF chromatin remodeling complex, and its degradation disrupts oncogenic transcriptional programs in synovial sarcoma cells, presenting a novel therapeutic avenue [9]. In addition to clinical-stage drug candidates, the table also includes preclinical PROTACs, such as a BCL-XL degrader for solid tumors and a LRRK2 degrader for Parkinson’s disease [10]. These early-stage therapies indicate a growing interest in expanding the scope of PROTACs beyond oncology into neurodegenerative and inflammatory diseases [10]. The extensive array of ongoing clinical trials highlights the significant potential of targeted protein degradation as a revolutionary strategy in drug development. In contrast to conventional small-molecule inhibitors, which frequently encounter issues such as off-target effects or the development of resistance, PROTACs present a mechanism-driven advantage by effectively eliminating pathogenic proteins [1,2]. However, challenges remain, including drug stability, bioavailability, and tissue-specific targeting [11]. Continued research and clinical validation will determine whether these emerging therapies can fulfill their promise as next-generation precision medicines [11].

### 2.2. Molecular Design and Action Mechanism of PROTACs Currently in Clinical Trials

The molecular design of PROTACs, currently in clinical trials, represents a sophisticated approach to targeted protein degradation, offering a novel mechanism for therapeutic intervention. PROTACs are hetero-bifunctional molecules composed of three key components (Figure 2A): a ligand that binds to the target protein, a ligand that recruits an E3 ubiquitin ligase, and a linker that connects these two ligands [1,2]. In addition to PROTACs, the field of targeted protein degradation is rapidly diversifying with the emergence of new modalities, such as LYTACs, AUTACs, DUBTACs, and molecular glues. These technologies extend the reach of degradation strategies to extracellular, membrane-bound, and even aggregated proteins, which were previously considered inaccessible to traditional small molecules. For example, LYTACs enable the selective lysosomal degradation of cell-surface receptors, while AUTACs harness the autophagy pathway for the removal of pathogenic proteins and damaged organelles. The integration of these modalities into clinical development pipelines is expected to open new therapeutic avenues for diseases that remain untreatable with current approaches (Figure 2B).

The rational design of PROTACs is a multifaceted process that integrates structural, biochemical, and pharmacokinetic parameters to enhance their therapeutic potential. Several key factors must be considered in the molecular design of PROTACs to optimize their efficacy and clinical applicability [1,2].

Binding affinity and selectivity are paramount considerations in the development of PROTACs, as they ensure robust and specific interactions with both the target protein and the recruited E3 ubiquitin ligase. The identification of a high-affinity ligand for the target protein is crucial for promoting efficient degradation, while the ligand for the E3 ligase must effectively engage the ubiquitin–proteasome machinery. Achieving optimal selectivity mitigates off-target effects and potential toxicity, thereby improving the therapeutic index [3]. The design of the linker is another crucial aspect, as it influences the efficiency of ubiquitination and subsequent degradation of the target protein. Linker length, rigidity, and flexibility must be fine-tuned to position the target protein and E3 ligase in close proximity while maintaining the overall stability of the PROTAC molecule. An optimally designed linker should support efficient ubiquitin transfer while ensuring adequate metabolic stability and cellular permeability [4]. Computational modeling and structure–activity relationship studies have been extensively employed to iteratively refine linker architecture, optimizing the overall pharmacological performance of PROTACs [5]. Pharmacokinetics and bioavailability play a significant role in the clinical success of PROTACs, as their relatively large molecular weight and physicochemical properties can impact absorption, distribution, metabolism, and excretion [6]. Strategies to enhance their bioavailability include modifications to physicochemical properties, such as lipophilicity and polarity; the development of prodrugs; and nanoparticle-based formulations to improve systemic circulation time and tissue targeting [7]. Despite these advances, oral bioavailability remains a major challenge, necessitating alternative delivery strategies, including parenteral administration routes [8]. The choice of E3 ligase is a determinant of degradation efficiency and tissue specificity. The most commonly utilized E3 ligases in PROTAC design are cereblon (CRBN) and von Hippel–Lindau (VHL), though newer ligases, such as MDM2, DCAF15, and Keap1, are being explored to broaden the therapeutic applicability of targeted protein degradation [9]. The selection of an E3 ligase that is highly expressed in the target tissue enhances the specificity of degradation while reducing systemic toxicity [10]. Expanding the repertoire of ligases used in PROTAC development may provide new opportunities to overcome resistance mechanisms and improve the clinical viability of these molecules [11]. By integrating these molecular design principles, researchers aim to refine the next generation of PROTACs to ensure improved efficacy, safety, and clinical applicability across diverse disease indications.

### 2.3. Current Clinical Trials

PROTACs have rapidly progressed into clinical evaluation, with multiple candidates currently under investigation across various therapeutic areas, including oncology, hematologic malignancies, and autoimmune diseases [12]. These clinical trials aim to establish the safety, efficacy, and therapeutic feasibility of targeted protein degradation in different patient populations (Figure 3). The past five years have witnessed a significant evolution in the clinical development of targeted protein degraders, particularly PROTACs. Key milestones from 2019 to 2024 are summarized in Figure 3, illustrating the transition from first-in-human studies to late-phase clinical trials and the expansion of this technology into new therapeutic areas. Looking ahead, the field of targeted protein degradation is poised for further breakthroughs. Ongoing and upcoming trials will determine the clinical impact of these agents in broader patient populations and new disease areas. Furthermore, the emergence of next-generation modalities, such as LYTACs and molecular glues, promises to further expand the therapeutic landscape, enabling the targeting of previously inaccessible proteins and pathways. Continued innovation in molecular design, clinical strategy, and combination therapies will be key to realizing the full potential of this transformative technology.

In oncology, PROTACs targeting key oncogenic drivers have demonstrated promising clinical outcomes. ARV-110, developed by Arvinas, is a first-in-class androgen receptor degrader currently in Phase 2 trials for metastatic castration-resistant prostate cancer (mCRPC) [13]. ARV-110 is designed to overcome resistance mechanisms associated with traditional androgen receptor inhibitors such as enzalutamide and abiraterone, offering an alternative therapeutic strategy for patients with treatment-resistant disease. Another advanced PROTAC candidate, ARV-471, is an estrogen receptor degrader being developed by Arvinas in collaboration with Pfizer. Currently in Phase 3 trials for estrogen receptor-positive, HER2-negative breast cancer, ARV-471 has demonstrated a superior ability to degrade estrogen receptor compared to conventional selective estrogen receptor degraders, such as fulvestrant, potentially improving treatment outcomes in hormone therapy-resistant patients [14].

Hematologic malignancies also represent a significant area of investigation for PROTAC-based therapies. CFT7455, developed by C4 Therapeutics, is a potent degrader of Ikaros (IKZF1) and Aiolos (IKZF3), two transcription factors critical for multiple myeloma cell survival. Currently in Phase 1 trials, CFT7455 is being evaluated for its efficacy in patients with relapsed or refractory multiple myeloma [15]. This approach may provide an alternative to immunomodulatory drugs such as lenalidomide by inducing complete degradation of IKZF1/3, potentially enhancing therapeutic efficacy in patients resistant to existing treatments [16].

Beyond oncology, PROTACs are being explored for their potential in autoimmune diseases. KT-474, a PROTAC developed by Kymera Therapeutics, is currently in Phase 1 trials for autoimmune conditions, including atopic dermatitis and hidradenitis suppurativa [17]. KT-474 targets IRAK4, a kinase involved in Toll-like receptor (TLR) and interleukin-1 receptor (IL-1R) signaling, which are critical mediators of inflammatory responses. By degrading IRAK4, KT-474 offers a novel mechanism for controlling inflammation without the broad immunosuppressive effects associated with traditional therapies [18]. The evolving clinical landscape of PROTACs underscores their potential to revolutionize the treatment paradigm for a variety of diseases. Ongoing clinical trials are anticipated to yield essential insights into the efficacy and safety of targeted protein degradation, thereby facilitating the advancement of next-generation therapies characterized by improved selectivity and durability.

The molecular design of PROTACs represents a transformative approach in drug discovery, leveraging the ubiquitin–proteasome system to achieve targeted protein degradation [19]. Unlike traditional small-molecule inhibitors, which merely block protein function, PROTACs induce the selective elimination of pathogenic proteins, offering advantages such as prolonged therapeutic effects and the ability to target previously undruggable proteins [20].

Despite their promise, several challenges remain, including optimizing pharmacokinetics, improving cellular permeability, and expanding the range of E3 ligases to enhance tissue specificity [21]. The potential for resistance mechanisms also warrants further investigation, particularly in oncology, where cancer cells may adapt by downregulating recruited E3 ligases or acquiring mutations that affect PROTAC binding [22]. Future research efforts will focus on refining PROTAC chemistry, optimizing delivery strategies, and exploring combination therapies to enhance their clinical effectiveness [23].

As clinical trials progress, PROTACs hold the potential to revolutionize targeted therapy across multiple disease areas, providing novel treatment options for patients with conditions that currently lack effective therapeutic alternatives [24]. The continued advancement of PROTAC technology will be instrumental in shaping the future of precision medicine, enabling the selective degradation of disease-driving proteins and expanding the therapeutic landscape for complex and treatment-resistant diseases [25].

## 3. Discussion

Recent clinical trials have significantly broadened the scope of PROTAC (Proteolysis-Targeting Chimera) applications, extending beyond oncology and into areas such as autoimmune diseases, neurodegenerative disorders, and hematologic malignancies. The rapid evolution of targeted protein degradation (TPD) reflects major advancements in molecular design, pharmacokinetics, and clinical translation, paving the way for novel therapeutic interventions in diseases that were previously intractable [3,12].

### 3.1. Expansion of PROTACs Beyond Oncology and Neurological Diseases

Initially developed for cancer treatment, PROTACs are currently being investigated for a diverse array of disease indications, encompassing autoimmune and neurodegenerative disorders. Their capacity to degrade previously “undruggable” proteins has generated considerable interest in their therapeutic potential for conditions marked by protein aggregation or dysregulated signaling pathways [14].

In the field of autoimmune diseases, KT-474, developed by Kymera Therapeutics, is currently in Phase 1 trials as a PROTAC targeting IRAK4. IRAK4 is a key mediator in Toll-like receptor (TLR) and IL-1 receptor (IL-1R) signaling, both of which are critical to inflammatory responses [14]. This candidate is being investigated for the treatment of atopic dermatitis and hidradenitis suppurativa, demonstrating the potential of PROTACs in modulating immune system function and inflammatory processes.

PROTACs have also gained traction in hematologic malignancies. NX-2127, developed by Nurix Therapeutics, is an oral BTK degrader currently under evaluation in Phase 1 clinical trials for B-cell malignancies, including chronic lymphocytic leukemia (CLL) and non-Hodgkin’s lymphoma (NHL) [15]. Unlike traditional BTK inhibitors such as ibrutinib, NX-2127 induces irreversible degradation of BTK, offering a promising strategy to overcome resistance mechanisms associated with BTK inhibitors. Additionally, BCL-xL PROTACs are in early-stage clinical development for hematologic malignancies, particularly leukemia and lymphoma [16]. The agents described herein are specifically engineered to enhance apoptotic sensitivity in malignancies that have acquired resistance to standard therapeutic interventions, thereby broadening the therapeutic scope of targeted protein degradation.

In neurodegenerative diseases, PROTACs are being explored for their ability to selectively degrade proteins implicated in neuronal dysfunction. DNL151 (BIIB122), a LRRK2 degrader developed by Denali and Biogen, is currently in Phase 2 clinical trials for Parkinson’s disease [17]. Mutations in LRRK2 are known to contribute to dopaminergic neuronal degeneration, and targeted degradation of this protein holds promise for slowing disease progression in genetically defined Parkinson’s patients. Moreover, PROTACs targeting tau and α-synuclein are in preclinical development for Alzheimer’s disease and multiple sclerosis [18]. These PROTACs aim to eliminate toxic protein aggregates that drive neurodegeneration, thereby potentially altering disease progression and improving patient outcomes.

The expansion of PROTACs into neurological and inflammatory disorders underscores their broad therapeutic potential. The capacity to target intracellular proteins implicated in neurodegeneration positions PROTACs as a promising strategy for addressing proteinopathies, while their application in autoimmune diseases indicates potential for immune modulation and inflammation control.

Beyond oncology and neurological disorders, the utilization of PROTACs is swiftly broadening into additional therapeutic domains. Recent preclinical and early clinical investigations have highlighted the efficacy of PROTACs in autoimmune conditions, including rheumatoid arthritis and systemic lupus erythematosus, by targeting critical signaling proteins associated with inflammation. Moreover, there is an increasing interest in employing PROTACs for infectious diseases, with initiatives aimed at degrading viral or bacterial proteins vital for pathogen survival. These advancements emphasize the versatility of PROTAC technology and its potential to meet unmet medical needs across a wider array of diseases.

### 3.2. Challenges in PROTAC Development

Despite their therapeutic potential, PROTACs encounter substantial challenges concerning pharmacokinetics, bioavailability, and resistance mechanisms [19,20]. Addressing these factors is essential to optimize their clinical utility and ensure sustained therapeutic success. A primary challenge in the development of PROTACs is their pharmacokinetic profile. Numerous PROTAC molecules demonstrate elevated molecular weight and limited cell permeability, which can restrict oral bioavailability and systemic distribution [21]. Enhancing linker chemistry is a vital strategy for improving cellular uptake, while the exploration of nanoparticle-based formulations is underway as a method for targeted drug delivery [22]. Additionally, alternative administration routes, such as intravenous and intranasal delivery systems, are being investigated to augment bioavailability and improve therapeutic outcomes [23].

Another critical factor affecting PROTAC efficacy is the selection of an appropriate E3 ligase to facilitate targeted protein degradation. Presently, PROTACs predominantly utilize cereblon (CRBN) and von Hippel–Lindau (VHL) ligases [24]. However, broadening the spectrum of E3 ligases employed in PROTAC design could enhance tissue specificity and degradation efficiency. The discovery of novel E3 ligases, including Mdm2, DCAF15, and Keap1, represents a promising direction for expanding the applicability of PROTACs across various disease contexts [25]. A more diverse repertoire of E3 ligases could assist in mitigating resistance mechanisms and enable more precise degradation of disease-associated proteins.

Resistance to PROTAC therapy has emerged as a potential limitation, particularly in oncology [26]. Some cancers adapt to PROTAC treatment by downregulating the recruited E3 ligase, thereby preventing efficient target protein degradation. Others acquire mutations in the target protein-binding sites, rendering the PROTAC molecule ineffective. Several strategies are being explored to overcome these resistance mechanisms [27]. The development of dual-targeting PROTACs, which degrade multiple oncogenic proteins simultaneously, could provide a means of bypassing resistance pathways. Combination-therapy approaches integrating PROTACs with kinase inhibitors or immunotherapies are also being investigated to prevent tumor adaptation and enhance treatment durability [28].

Addressing these challenges is essential to realizing the full potential of PROTAC-based therapies. Advancements in linker chemistry, the identification of alternative delivery platforms, and continued exploration of novel E3 ligases will be instrumental in improving PROTAC efficacy, bioavailability, and therapeutic durability [29].

To overcome the challenges associated with PROTAC development, several strategies are being actively pursued. Improving oral bioavailability and metabolic stability can be achieved through rational modification of physicochemical properties; prodrug approaches; or advanced drug delivery systems, such as nanoparticles. Enhancing selectivity and minimizing off-target effects may involve the use of tissue-specific E3 ligases or the design of PROTACs with optimized linker lengths and rigidity. Resistance mechanisms can be addressed by expanding the repertoire of E3 ligases and developing dual-target PROTACs that degrade multiple disease drivers simultaneously. Ongoing research in these areas is expected to further refine the clinical potential of PROTACs and related technologies.

## 4. Conclusions

### 4.1. PROTACs Clinical Landscape

Targeted protein degradation signifies a paradigm shift in assumptions regarding drug development, presenting a novel strategy to selectively eliminate pathogenic proteins rather than merely inhibiting their function. The growing number of PROTACs entering clinical trials in oncology, autoimmune diseases, and neurodegenerative disorders underscores their transformative potential [30]. By utilizing the ubiquitin–proteasome system to degrade disease-causing proteins, PROTACs offer an innovative therapeutic mechanism with extensive applicability across various disease areas.

The rapid evolution of PROTACs from a novel concept to clinical reality underscores their transformative potential in modern therapeutics. Initially focused on oncology, PROTACs are now expanding into diverse disease areas, including infectious diseases [31], autoimmune disorders, and neurodegenerative conditions, demonstrating their versatility in targeting previously “undruggable” proteins. Advances in E3 ligase recruitment, such as the exploitation of non-canonical ligases [32], and innovative delivery systems, like antibody–PROTAC conjugates [33], address longstanding challenges in bioavailability and tissue specificity. Meanwhile, computational tools, including machine learning for ternary complex optimization [34], are accelerating rational design.

Despite these breakthroughs, hurdles remain, particularly in overcoming resistance mechanisms [35] and refining pharmacokinetics for broader applications, such as cardiovascular diseases [36]. The clinical success of ARV-110 in prostate cancer [37] and the emergence of alternative degradation modalities (e.g., LYTACs) highlight the dynamic landscape of targeted protein degradation [38]. As PROTACs advance toward regulatory approval, their integration with precision-medicine platforms is poised to redefine treatment paradigms across various therapeutic areas. Future initiatives must emphasize combinatorial strategies, E3 ligase diversification, and patient stratification to fully harness this potential, ultimately revealing a new era of disease-modifying therapies. Future research should concentrate on optimizing pharmacokinetic properties, broadening the spectrum of E3 ligases to enhance specificity and efficacy, and addressing potential resistance mechanisms. If these challenges can be effectively addressed, PROTACs could emerge as a cornerstone of precision medicine, fundamentally transforming therapeutic strategies for complex and previously untreatable diseases.

### 4.2. Comparison of PROTACs with Other Degrader Approaches

While the primary focus of this review is on PROTACs currently in clinical development, we recognize the importance of other targeted protein-degradation technologies (Figure 2B). To provide a more balanced perspective, we have expanded our discussion to include emerging modalities such as LYTACs (lysosome-targeting chimeras), molecular glues, and AUTACs (autophagy-targeting chimeras). These alternative approaches utilize distinct mechanisms—such as lysosomal or autophagic degradation pathways—and are beginning to enter early clinical evaluation. By comparing these modalities with PROTACs, we aim to offer a comprehensive overview of the evolving landscape of protein-degrading therapeutics (Table 2).

In more detail, it was demonstrated that LYTACs could direct otherwise inaccessible proteins to the lysosome for degradation by engaging cell-surface lysosome-targeting receptors. This work significantly expanded the scope of targeted protein degradation beyond the cytosolic and nuclear proteins accessible to PROTACs [42]. Recent studies further advanced the LYTAC platform by engineering molecules that exploit the asialoglycoprotein receptor (ASGPR) for targeted lysosomal degradation of membrane proteins. Their findings validated the feasibility of LYTACs in live cells and highlighted the potential for targeting disease-associated cell-surface proteins, thereby broadening the therapeutic applications of targeted protein degradation [43].

In the case of molecular glues, a groundbreaking study revealed the mechanism by which the immunomodulatory drug lenalidomide acts as a “molecular glue” to promote the cereblon-dependent ubiquitination and degradation of transcription factors IKZF1 and IKZF3 [44]. The discovery provided a mechanistic basis for the clinical efficacy of lenalidomide in multiple myeloma and established the concept of molecular glues as small molecules that induce or stabilize protein–protein interactions for targeted degradation. A recent perspective article also discussed the rise of molecular glues as a new paradigm in drug discovery, further highlighting their therapeutic potential [45].

Furthermore, recent papers described AUTACs, a novel class of small molecules that tag target proteins for selective autophagic degradation. Their study demonstrated that AUTACs could induce the degradation of specific proteins and even damaged mitochondria by leveraging the autophagy–lysosome pathway, providing a new approach for the removal of pathogenic proteins and organelles [47]. Recent findings expanded on the AUTAC concept, describing the design and mechanism of AUTACs in more detail. Their review summarizes recent advances in autophagy-targeting strategies, discusses the potential therapeutic applications of AUTACs, and highlights the challenges and opportunities in harnessing autophagy for targeted protein degradation [48].

The field of targeted protein degradation has undergone a remarkable transformation in recent years, with PROTACs and other novel modalities, such as LYTACs, molecular glues, and AUTACs, reshaping the landscape of therapeutic development. These innovative approaches offer the unprecedented ability to eliminate disease-driving proteins, including those previously deemed “undruggable,” and have demonstrated promising efficacy across a broad spectrum of diseases, from oncology and hematologic malignancies to autoimmune and neurodegenerative disorders.

### 4.3. Final Considerations

Clinical trials of first-generation PROTACs, including ARV-471, ARV-110, CFT7455, and KT-474, have provided encouraging early results, highlighting the potential of this technology to overcome resistance mechanisms and deliver durable clinical responses. The rational design of these molecules—incorporating optimal target and E3 ligase selection, linker engineering, and strategies to enhance bioavailability—continues to evolve, addressing many of the initial challenges related to stability, specificity, and pharmacokinetics. At the same time, the emergence of alternative protein-degradation platforms, such as LYTACs and molecular glues, is expanding the therapeutic reach of targeted degradation to extracellular, membrane, and otherwise intractable proteins. As these modalities progress from preclinical studies to early clinical evaluation, they promise to further diversify the arsenal of precision medicines available.

Despite these advances, significant challenges remain. Achieving tissue-specific degradation, minimizing off-target effects, and optimizing delivery methods are active areas of research that will determine the long-term success and safety of these therapies. Moreover, a deeper understanding of resistance mechanisms and the development of next-generation degraders—potentially leveraging novel E3 ligases or dual-targeting strategies—will be critical for broadening the clinical impact of this technology. In summary, targeted protein degradation represents a paradigm shift in drug discovery and development. The ongoing progress in clinical trials, coupled with advances in molecular design and the emergence of new degradation modalities, positions this field at the forefront of precision medicine. Continued interdisciplinary collaboration and innovation will be essential to realize the full therapeutic potential of these agents, ultimately transforming the treatment landscape for patients with complex and refractory diseases.

Recent advances in artificial intelligence (AI) and machine learning are accelerating the discovery and optimization of protein degraders. AI-driven platforms can predict optimal linker length, E3 ligase compatibility, and target engagement profiles, significantly reducing the time required for lead optimization [49]. These computational approaches are also being utilized to identify novel E3 ligase recruiters and to design degraders with improved pharmacokinetic and pharmacodynamic properties. As these tools become more sophisticated, they are poised to revolutionize the rational design of next-generation degraders [50,51].

Furthermore, there is growing interest in combining PROTACs with other therapeutic modalities, such as immune checkpoint inhibitors, kinase inhibitors, or conventional chemotherapies [52]. Preclinical studies suggest that targeted protein degraders can sensitize tumors to immunotherapy by modulating the tumor microenvironment or by depleting immune evasion factor. In more detail, recent studies demonstrated that a PROTAC targeting CDK9 achieved more sustained and complete suppression of oncogenic transcription than traditional CDK9 inhibitors [53]. Their selective CDK9 degrader led to greater downregulation of MYC and anti-apoptotic proteins, resulting in enhanced cancer cell death. This work highlights how degradation can surpass inhibition for certain targets, supporting the therapeutic promise of PROTACs.

Additionally, PROTACs may overcome resistance mechanisms that limit the efficacy of standard treatments, providing a rationale for combination regimens in future clinical trials. ARD-69 was developed recently in that context. It is a highly potent PROTAC that induces rapid and near-complete degradation of the androgen receptor in prostate cancer models. ARD-69 showed superior efficacy compared to existing AR inhibitors, including activity against resistant cancer cells [54]. This study illustrates the potential of rational PROTAC design to overcome drug resistance and improve therapeutic outcomes in prostate cancer. The ability of targeted protein degraders to eliminate disease-causing proteins, rather than simply inhibit their activity, represents a paradigm shift in drug development [1]. This approach holds particular promise for patients with refractory or relapsed disease, rare genetic disorders, and conditions driven by proteins lacking enzymatic activity. As clinical experience with these agents grows, it is anticipated that protein degraders will become an integral part of precision medicine, offering new hope to patient populations with limited treatment options

## Figures and Tables

**Figure 1 pharmaceutics-17-00744-f001:**
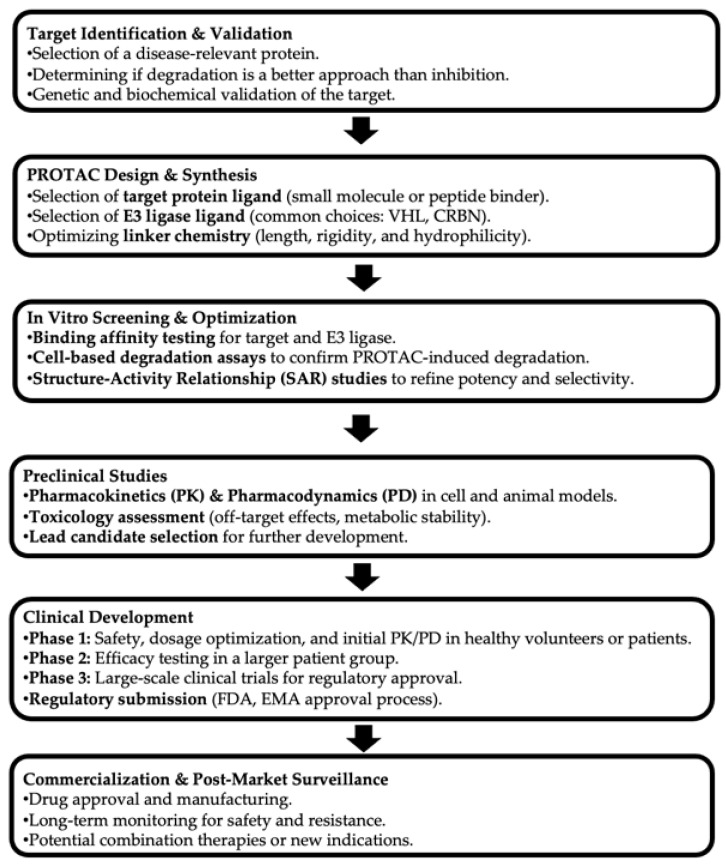
Drug development pipeline of PROTACs. (PROTAC: Proteolysis Targeting Chimera. HL: Von Hippel-Lindau. CRBN: Cereblon. FDA: Food and Drug Administration. EMA: European Medicines Agency).

**Figure 2 pharmaceutics-17-00744-f002:**
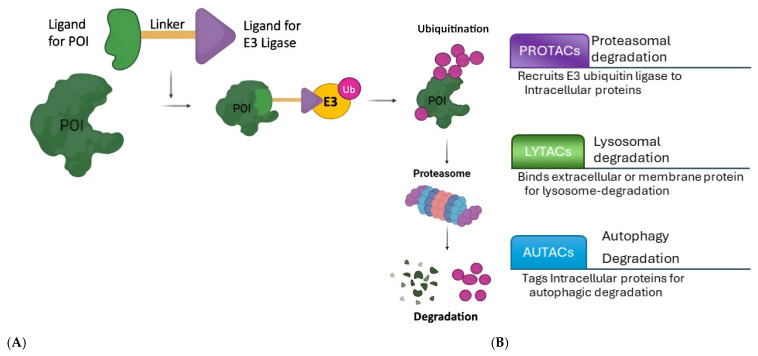
(**A**) Action mechanism of the PROTACs (POI: protein of interest, E3: E3 ubiquitin ligase; Ub: ubiquitin). (**B**) Comparison of PROTACs (Proteolysis-Targeting Chimeras) with LYTACs (lysosome-targeting chimeras) and AUTACs (autophagy-targeting chimeras).

**Figure 3 pharmaceutics-17-00744-f003:**
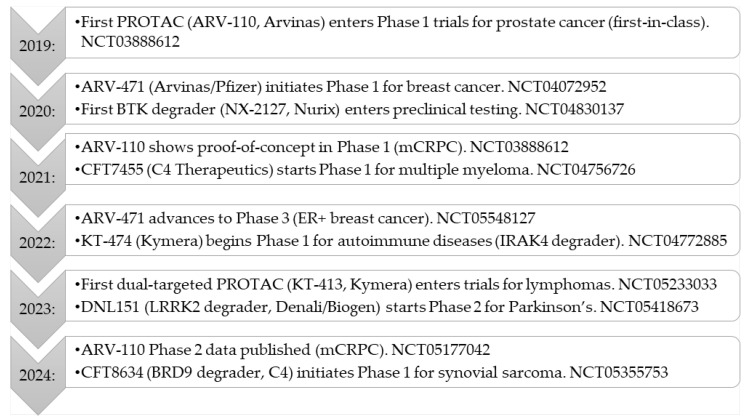
PROTACs clinical milestones (2019–2024).

**Table 1 pharmaceutics-17-00744-t001:** PROTACs that are currently being clinically tested [1,2,3,4,5].

Drug Name	Target	Indication	Company	Clinical Trial Phase
ARV-471	Estrogen receptor (ER)	Metastatic breast cancer	Arvinas/Pfizer	Phase 3, NCT05548127, NCT04072952
ARV-110	Androgen receptor (AR)	Prostate cancer	Arvinas	Phase 2, NCT03888612, NCT05177042
ARV-766	Androgen receptor (AR)	Prostate cancer	Arvinas	Phase 1/2, NCT05067140
CFT7455	IKZF1/3	Multiple myeloma	C4 Therapeutics	Phase 1, NCT04756726
KT-474	IRAK4	Autoimmune diseases	Kymera Therapeutics	Phase 1, NCT04772885
CC-90009	GSPT1	Acute myeloid leukemia (AML)	Celgene (Bristol-Myers Squibb)	Phase 1, NCT02848001
BCL-XL degrader	BCL-XL	Solid tumors	Not specified	Preclinical
NX-2127	Bruton’s Tyrosine Kinase (BTK)	B-cell malignancies	Nurix Therapeutics	Phase 1, NCT04830137
ACBI-001	BCL6	Non-Hodgkin lymphoma	Accent Therapeutics	Phase 1, NCT05508943
DT2216	BCL-XL	Hematologic malignancies	Dialectic Therapeutics	Phase 1, NCT05233033
KT-413	IRAK4/IMiD substrates	Lymphomas	Kymera Therapeutics	Phase 1, NCT05355753
CFT8634	BRD9	Synovial sarcoma	C4 Therapeutics	Phase 1, NCT05355753

**Table 2 pharmaceutics-17-00744-t002:** Comparison of clinical statuses for PROTACs (Proteolysis-Targeting Chimeras), LYTACs (lysosome-targeting chimeras), AUTACs (autophagy-targeting chimeras), and molecular glues.

Modality	Mechanism of Action	Clinical Status	References
PROTACs	Recruit E3 ubiquitin ligase to target protein, leading to ubiquitination and proteasomal degradation	Multiple agents in Phase 1–3 clinical trials (e.g., ARV-471, ARV-110, CFT7455, and KT-474)	[39,40,41]
LYTACs	Recruit lysosome-targeting receptors to mediate lysosomal degradation of extracellular and membrane proteins	Preclinical, early clinical evaluation	[42,43]
Molecular glues	Induce or stabilize protein–protein interactions between E3 ligase and target, leading to ubiquitination and degradation	Some agents (e.g., lenalidomide and CC-885) approved or in clinical trials	[44,45,46]
AUTACs	Tag target proteins with autophagy-targeting motifs for selective autophagic degradation	Preclinical	[47,48]

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
