# Peer review of "Molecular Design of Novel Protein-Degrading Therapeutics Agents Currently in Clinical Trial"

_pharmaceutics, 2025, doi:10.3390/pharmaceutics17060744_

Round 1

Reviewer 1 Report

Comments and Suggestions for Authors

This review provides a timely and comprehensive overview of PROTACs in clinical development. The authors effectively summarize key PROTAC candidates, their mechanisms, and ongoing trials across multiple therapeutic areas. The manuscript is well-organized and clearly written, with a strong emphasis on translational potential. However, some sections could benefit from additional refinement to enhance clarity, balance, and depth. For example, the title does not reflect the PROTAC-centric focus of the review. The title is overly broad and could mislead readers into assuming the review covers all protein degradation technologies (e.g., PROTACs, LYTACs, molecular glues, AUTACs). However, the manuscript focuses almost exclusively on PROTACs, with only a passing mention of LYTACs in the conclusion. A reader might expect a discussion of LYTACs (lysosome-targeting chimeras) or other degraders in clinical trials, but these are not covered in depth. 

Author Response

Comment:

This review provides a timely and comprehensive overview of PROTACs in clinical development. The authors effectively summarize key PROTAC candidates, their mechanisms, and ongoing trials across multiple therapeutic areas. The manuscript is well-organized and clearly written, with a strong emphasis on translational potential. However, some sections could benefit from additional refinement to enhance clarity, balance, and depth. For example, the title does not reflect the PROTAC-centric focus of the review. The title is overly broad and could mislead readers into assuming the review covers all protein degradation technologies (e.g., PROTACs, LYTACs, molecular glues, AUTACs). However, the manuscript focuses almost exclusively on PROTACs, with only a passing mention of LYTACs in the conclusion. A reader might expect a discussion of LYTACs (lysosome-targeting chimeras) or other degraders in clinical trials, but these are not covered in depth. 

Response: Thank you very much for your thoughtful and constructive feedback on our manuscript. We appreciate your positive comments regarding the clarity, organization, and comprehensive coverage of PROTACs in clinical development.

In response to your suggestion, we have revised the manuscript to more accurately reflect its PROTAC-centric focus in the title and introductory sections. Additionally, we have expanded our discussion to include more detailed information on other targeted protein degradation technologies, such as LYTACs, as well as a brief overview of molecular glues and AUTACs. This added content aims to provide readers with a broader perspective on the current landscape and ongoing clinical development of these emerging approaches.

We hope these changes address your concerns and further enhance the clarity and value of our review. Thank you again for your insightful feedback.

Reviewer 2 Report

Comments and Suggestions for Authors

Unfortunately, the present form of this review article ("Molecular design of novel Protein-degrading therapeutics agents currently in Clinical trial", Manuscript ID: Pharmaceutics-3620995; by Kacin et al.) is quite weak and it requires extensive improvements. The authors have missed discussing a lot of crucial points. They also need to demonstrate why there is a need for a new review article on such a topic, on which several articles have already been published. They need to prove why and where this review article stands out compared to all the previously published reports. I, therefore, did not find the present form of the review article (quality-wise) suitable enough for publication in the Pharmaceutics journal.

Author Response

Comment: Unfortunately, the present form of this review article ("Molecular design of novel Protein-degrading therapeutics agents currently in Clinical trial", Manuscript ID: Pharmaceutics-3620995; by Kacin et al.) is quite weak and it requires extensive improvements. The authors have missed discussing a lot of crucial points. They also need to demonstrate why there is a need for a new review article on such a topic, on which several articles have already been published. They need to prove why and where this review article stands out compared to all the previously published reports. I, therefore, did not find the present form of the review article (quality-wise) suitable enough for publication in the Pharmaceutics journal.

Response: We acknowledge your point that the current version of our review may lack coverage of certain crucial aspects and does not sufficiently highlight its novelty in the context of existing literature. In response, we are undertaking a thorough revision of the manuscript to address these issues.

We are incorporating additional discussion on key recent advances and emerging challenges in the field of protein-degrading therapeutics, including comprehensive coverage of alternative modalities such as LYTACs, molecular glues, and AUTACs, as well as the latest clinical trial updates and mechanistic insights.

We are revising the introduction to clearly articulate the need for this review, emphasizing recent developments and clinical progress that have not been comprehensively covered in previous publications. We will also provide a comparative analysis to highlight how our review differs from and adds value to existing reports.

We are including a dedicated section that summarizes the unique aspects of our review, such as our focus on molecular design strategies, translational challenges, and the future outlook for clinical development, to demonstrate the distinct contribution of our work.

Reviewer 3 Report

Comments and Suggestions for Authors

Recommendation: Major revision 

Comments:

Authors need to address why this work is important in the current scenario.

How has this article addressed different and significant information as compared to the published works? Please address this in the introduction part.

The contents in 'Figure 1: Drug development pipline of PROTACs' are very small. Please update this. Also, check the spelling 'pipline'. It should be 'pipeline'.

How can the 'Challenges in PROTAC Development' be encountered? The authors can provide an outline.

The contents in the section 'Expansion of PROTACs Beyond Oncology and Neurological Diseases' need to be increased.

The authors should include the source from where they have collected the information presented.

A thorough grammar and typographical checking of the manuscript is required.

Comments on the Quality of English Language

A thorough grammar and typographical checking of the manuscript is required.

Author Response

We thank you very much for your detailed and constructive comments on our manuscript. We appreciate your valuable feedback and have carefully considered each of your suggestions. Below, we outline the revisions we have made in response:

Comments: Authors need to address why this work is important in the current scenario.

How has this article addressed different and significant information as compared to the published works? Please address this in the introduction part.

Revision: We have thoroughly revised the introduction to clearly articulate the importance of this review in the current scenario. Specifically, we emphasize recent advances in the clinical development of protein-degrading therapeutics, highlight emerging trends, and discuss gaps in the literature that our review addresses. We also provide a comparative summary of how our article offers new insights and updated information not covered in previous reviews.

Comment: The contents in 'Figure 1: Drug development pipline of PROTACs' are very small. Please update this. Also, check the spelling 'pipline'. It should be 'pipeline'.

Revision: We have updated Figure 1 to improve clarity and readability by enlarging the contents and refining the layout. The spelling error "pipline" has been corrected to "pipeline."

Comment: How can the 'Challenges in PROTAC Development' be encountered? The authors can provide an outline.

Review: We have expanded the section on challenges in PROTAC development to include a structured outline of potential strategies to overcome these obstacles, such as optimizing pharmacokinetics, improving cell permeability, and addressing resistance mechanisms.

Comment: The contents in the section 'Expansion of PROTACs Beyond Oncology and Neurological Diseases' need to be increased.

Review: We have significantly increased the content in this section to provide a more comprehensive overview of PROTAC applications in other therapeutic areas, including autoimmune diseases, infectious diseases, and rare disorders, supported by recent examples and clinical trial updates.

Comment: The authors should include the source from where they have collected the information presented.

Review: We have ensured that all data and statements are now appropriately referenced, and have included detailed citations for the sources of information presented throughout the manuscript.

Comment: A thorough grammar and typographical checking of the manuscript is required.

Review: The manuscript has undergone thorough grammatical and typographical review to improve the overall quality and readability.

We believe these revisions address your concerns and substantially enhance the quality and impact of our manuscript. Thank you again for your insightful comments, which have been invaluable in guiding our improvement

Round 2

Reviewer 2 Report

Comments and Suggestions for Authors

The authors have answered most of my comments and suggestions. They have done quite a bit of additional writing with proper literature support in response to my comments. As a result, the quality of the review article has improved significantly. I, therefore, recommend the review article for publication in the Pharmaceutics journal.

Reviewer 3 Report

Comments and Suggestions for Authors

I have reviewed the revised manuscript and found the authors have addressed most of these points.